# Re-imagining health research to include the voices of justice-impacted individuals

**Romina Foster-Bonds****, Julia Moore Vogel**[ID]*

Scripps Research Translational Institute, Scripps Research, La Jolla, California, United States of America

* juliamv@scripps.edu

## Abstract

People with prior justice system involvement experience health disparities, yet are often underrepresented in research. We sought to 1) explore the perspectives of individuals with prior justice involvement regarding research and science and 2) identify barriers to their participation in research. Focus groups with 67 participants from five advocacy groups across the United States revealed interest in research participation. To enable participation, researchers need to accommodate prior commitments and offer adequate compensation. Collaboration with advocacy organizations supporting justice-impacted individuals can facilitate recruitment and engagement goals. This study can inform optimal strategies for engaging individuals impacted by the justice system in research. The voices of justice-impacted individuals can provide diverse perspectives on health research programs and allow for the collection of meaningful data to understand the effects and possible solutions for severe health disparities in the United States.

## Introduction

The United States (US) incarcerates more people per capita than any other country; 4.9 million people in the US are formerly incarcerated and 79 million or more have a criminal record [1]. The increased incarceration rate in the US is not because crime has increased; crime rates have declined since the 1990s [2]. The arrest rate, particularly for non-violent drug crimes, has increased dramatically, while sentences have gotten longer [3,4]. Across 187 cities in the US, there is a significant increase in laws criminalizing behaviors related to homelessness such as bans on sleeping, sitting or lying down in public, sleeping in your car, begging and loitering [5], and 9 states have banned sharing food with homeless people [6]. Homelessness makes someone 11 times more likely to be incarcerated and being incarcerated makes someone 10 times more likely to be homeless [7].

With increasing incarceration, there are also growing disparities (Fig 1). Black men and women are five times more likely to be incarcerated than White men, and Hispanic men are 1.3 times more likely to be incarcerated than White men [8,9].

**Data availability statement:** Summarized data is available in the supplemental materials. In the IRB-approved protocol and participant communications we state the transcript will be destroyed upon completion of the study report. The original transcripts were de-identified upon initial analysis and destroyed once the study report was completed. The study report was utilized to write this publication, which includes the minimal dataset. We will ensure this data is stored for at least 7 years, with support from Scripps Research Information Technology team. Any requests for re-use of de-identified transcript will be reviewed by the All of Us Research Program IRB. Those interested in accessing the data can contact allison.mandich@nih.gov and juliamv@scripps.edu. Although the authors cannot make their study's data publicly available at the time of publication, all authors commit to make the data underlying the findings described in this study fully available without restriction to those who request the data, in compliance with the PLOS Data Availability policy. For data sets involving personally identifiable information or other sensitive data, data sharing is contingent on the data being handled appropriately by the data requester and in accordance with all applicable local requirements.

**Funding:** Both RFB and JMV's contributions to this study were funded by the Department of Health and Human Services; National Institute of Health; Office of The Director; All of Us Research Program (U24OD023176 and 1OT2OD035580-01). The funders provided strategic support and guidance on the study design and budget but had no role in the data collection, analysis, decision to publish or preparation of the manuscript.

**Competing interests:** The authors have declared that no competing interests exist.

Black men and Hispanic men are less likely to receive a probationary sentence compared to White men, at 23.4% and 26.6% respectively. Black women are 11.2% and Hispanic women are 29.7% less likely to receive a probation sentence compared to White women [10].

There are severe, immediate, and long standing racial and socioeconomic disparities for justice-impacted individuals compared to others, including negative effects on mental, physical, and social health which often significantly impairs successful post-incarceration societal reintegration. Post-incarceration, involvement with the justice system sanctions denying employment, government assistance, housing, and the right to vote. In the 2-weeks following release, individuals are at a 12-fold increased risk of death [11]. A report from the Bureau of Justice Statistics in 2010 found that out of over 50,000 people released from prison, 65% were not able to find employment over 4 years after their release [12]. They are at higher risk for injury and death than the general public and have disproportionately high rates (80%) of serious mental illness, substance use disorder, infections, and other chronic physical health conditions [13].

The negative effects of being justice-impacted often extend beyond the individual to their family members; 45% of people in the US have an immediate family member who is incarcerated and this is more prominent in Black families where 63% of Black individuals have an immediate family member who is incarcerated [14]. Furthermore, one in 14 White children, and one in 8 Black children, have had a parent go to jail or prison [15]. Children of incarcerated parents face immediate threats to their financial and emotional stability and, for many, there are severe consequences that result from lack of parental income and housing. Recent research has elucidated how biomarkers of chronic stress; such as accelerated telomere length shortening, allostatic load, and cortisol/cortisone concentration, may be associated with parental incarceration and specific incarceration-related traumas [16–18]. Parental incarceration is classified as an adverse childhood experience [19], impacting physical and mental health, developmental issues, and educational needs [20], and therefore has potential ramifications over generations of families.

The classification of individuals in prison as a "vulnerable population" has limited the research participation of justice-impacted individuals. This well-intentioned classification was implemented in response to harmful research conducted in US prisons [21]. However, regulations can restrict the right of incarcerated individuals to participate in ethical research designed to benefit them [22,23]. These restrictions have resulted in a growing gap of evidence that inhibits the understanding and reduction of health disparities within prison and after release [24]. Aside from regulatory challenges to research participation, there have been financial reasons for the lack of research related to incarceration or incarcerated individuals. A review of incarceration-related research between 1985 and 2022 found that only 0.12% of all projects funded at the NIH and 0.03% at the NSF were related to incarceration [25], which aligns with previous research on funding for criminal justice health research [26]. The amount of funding is nearly 100 times less than funding for research on education (9.38% of projects funded [25]), and in 2012 was only 1.5% of the NIH's

| Predictors | • Foster care, received poor school education, are indigenous, have unsupported MH and cognitive disability, come from or live in a disadvantaged location<br>• Homelessness makes someone 11 times more likely to be incarcerated | Effects | • 2x likely to have no high school diploma, and 8 times less likely to complete college<br>• >60% not able to find employment over 4 years after their release<br>• 10x likely to be homeless | • Inc risk injury and death Inc risk (80%) SMI, SUD and infectious and other chronic physical health conditions<br>• Parental incarceration is considered an adverse childhood experience; impacting physical and mental health, developmental issues, and educational needs of their children |
|---|---|---|---|---|
| | **Pre-incarceration** | | **Incarceration** | **Post-Incarceration** |
| Justice Impact | • The US incarcerates more people per capita than any other nation<br>• There is less probation for minorities. Despite less crimes since the 1990s, arrests have increased. Particularly for Black and Hispanic men who are less likely to receive probation for drug arrests<br>• And more laws criminalizing behaviors related to homelessness | | • More prison labor, less rehabilitation. Prison labor provides an average of <$5 a day for work, with some states where they receive no pay<br>• There is little to no research funding, estimated at 0.12% of NIH and 0.03% of NSF funding | • Denial of citizen's right to work, receive government assistance, housing and vote<br>• Inability to access treatment, limited EHR sharing from the prison system |

**Fig 1. Review of Multimodal and Intergenerational predictors and effects of Justice in the US.**

overall investment in health disparities research [26]. The National Academies of Sciences, Engineering and Medicine and scholars have called for the examination of mass incarceration as a possible hazardous exposure for individuals and their families [27–29]. With over 600,000 individuals released from state and federal prisons each year [30], public health initiatives and research programs can fill the gap in data related to incarceration while connecting individuals to a community that shares a common interest in improving their and their community's health.

The *All of Us* Research Program (*All of Us*) has created a large-scale, inclusive health research database [31]. A focus of *All of Us* is ensuring participation is available to all interested and eligible individuals in the US. This is achieved by actively engaging historically underrepresented communities [32]. Individuals with previous US justice system involvement are one of the largest and most underserved populations in the US. We conducted focus groups in collaboration with five advocacy organizations to help *All of Us* improve understanding of perspectives of justice-impacted individuals on science, participating in research, as well as facilitators and barriers to participation.

## Materials and methods

### Recruitment

Participants were recruited by five partnering advocacy organizations in the US: The Dannon Project in Birmingham Alabama, The Center for Returning Citizens in Philadelphia Pennsylvania, REVOLVE in Wilmington Delaware, Center For Employment Opportunity in New York City New York and Coalition On Adult Basic Education in Bradenton Florida (See Table 1). The advocacy organizations had existing relationships with study participants, as these organizations provide services to support these individuals. Inclusion criteria were: individuals who have previously been incarcerated, are not currently incarcerated, and are at least 18 years old. Recruitment of 67 participants was done by various methods depending on the preference of the advocacy partner including; phone calls, emails, social media posts and text messages (S1 Text for details). Participation was voluntary.

**Table 1. Advocacy Organization Partners.**

| Organization | Description | Location and Website |
|---|---|---|
| Revolve Staffing and Training | A full staffing agency with an in-house Training program, that focuses on the growth of good citizens and a true passion for the Re-Entry community. | Wilmington, DE Website |
| The Center for Returning Citizens | Helps persons who are returning to society to establish themselves and move forward in a positive manner. | Philadelphia, PA Website |
| The Dannon Project | Ensures maximum utilization of private charitable resources and government funding to support the development of healthy communities and lifestyles to decrease elevated-risk behaviors and economic exclusion caused by generational cycles of poverty. | Birmingham, AL Website |
| Center For Employment Opportunity (CEO) | Works to reduce recidivism and increase employment by providing people returning from prison immediate paid employment, skills training, and ongoing career support. To offer work experience, CEO operates transitional work crews that provide supplemental indoor/outdoor maintenance and neighborhood beautification services to more than 40 customers across the U.S. | New York, NY Website |
| Coalition On Adult Basic Education | Inspires educators so adults succeed and communities thrive. We provide leadership, professional development, advocacy, and communication services that encourage greater consciousness and cultural competency in our interactions with teachers, administrators, adult learners, and our partners. We are committed to using our platform and influence to celebrate, engage with, and listen to all adult education communities and diverse voices of our field. | Bradenton, FL Website |

## Study design

Focus groups were conducted by Adachi-Odai Solutions, a research logistics company based in Alabama, using an IRB approved script (S2 Text). Each advocacy organization had two focus groups that included only the participants they recruited. Focus groups were scheduled in collaboration with the recruitment partner and implemented on Zoom to utilize mobile app and phone call in features as well as participant video off controls to preserve anonymity. Each interview lasted about 75–120 minutes. If participants did not have access to a phone or computer for the Zoom meeting, the partnering advocacy organizations provided a location and equipped participants with a computer or tablet for their participation in the focus group. For participants with access to a phone or computer but a limited understanding of joining the zoom (by phone or computer) and using zoom features such as mute, the partnering advocacy organization held short training sessions where they practiced joining and utilizing the zoom call features prior to the scheduled focus group. Participants were reminded how to use the zoom features and were supported if they had any issues or questions for how to use the features. As of 2023, 90% of adults in the US owned smartphones, including 91% of Hispanic adults, 84% of Black adults, and 79% of people with annual incomes below $30,000 [33]. This provided participants the flexibility to join from their place of choice, with no transportation requirement. It also facilitated anonymity since they did not have to present in person. Participants were given the following information: the moderator's name and role, that Scripps Research was conducting the study, the study objectives and measures the study took to maintain the participant's privacy (e.g., destruction of transcripts after completion of the study report, use of pseudonyms). This was followed by an ice breaker and ground rules for the focus group. The moderator then began to ask questions related to each objective, aiming to create opportunities for as many individuals to answer the questions and share their thoughts while also attempting to reach as many questions as possible across objectives. Participants were told they would receive a list of resources from the partnering advocacy organization and receive an email with a $70 electronic gift card within 72 hours of the focus group as a form of compensation for their time. Advocacy partner staff collected demographic data and aimed to reflect the Returning Citizens population (View Table 2).

## Transcription

All audio recordings were transcribed verbatim using Zoom's transcription tool to ensure that every detail, including pauses and inflections, was captured, providing a complete textual representation of the discussions.

**Table 2. Participant Demographics.**

| Demographics | Number of participants | Percent of total |
|---|---|---|
| **Age** | | |
| 18-25 | 9 | 13% |
| 26-40 | 33 | 49% |
| 41-54 | 18 | 27% |
| 55+ | 8 | 12% |
| **Race** | | |
| White | 8 | 12% |
| Black | 42 | 63% |
| Middle Eastern | 1 | 1% |
| Hispanic | 10 | 15% |
| Mixed | 3 | 4% |
| Native Hawaiian or Other Pacific Islander (NHPI) | 1 | 1% |
| American Indian or Alaska Native (AIAN) | 2 | 3% |
| **Gender** | | |
| Male | 48 | 72% |
| Female | 19 | 28% |
| **Household Income** | | |
| under $10,000 | 11 | 16% |
| $10,000-$24,999 | 24 | 36% |
| $25,000-$34,999 | 8 | 12% |
| $35,000-$49,999 | 10 | 15% |
| $50,000+ | 14 | 21% |

The transcriptions were reviewed by the focus group moderator against the audio recordings for accuracy [34,35]. Any discrepancies, inaudible sections, or misinterpretations were corrected, resulting in high-quality transcripts suitable for qualitative analysis [34].

## Risks and mitigation

To protect privacy, the advocacy partners collected demographic information verbally and entered the data into a secure, password protected google form (S1 Table) in a secure cloud folder saved within the Scripps Research database. Participant phone numbers were kept only by the Advocacy partners in their database and were not combined with the study data or shared with Scripps for the study. The databases of advocacy partners were either (1) online through their website which has a CRM (customer relationship management) system and is password protected, or (2) paper applications kept in a locked filing cabinet.

Each participant selected a pseudonym to use and any mention of personally identifying information was removed from the transcripts. During the focus groups, participants were unable to turn on their video and were addressed and identified by their chosen pseudonyms, ensuring that their real identities remained safeguarded within the group. Although participants' email addresses were saved to provide gift cards to the participants, this information was not linked to participants' responses and were deleted from Scripps servers once the compensation process was complete.

To mitigate stigmatization and re-identification risks, our findings focus on overarching themes instead of specific details or case studies. Findings are framed to align with the study's specific aims, avoiding reinforcement of stereotypes or negative perception.

## Data analysis

The focus group data were analyzed using an applied Thematic Analysis (TA) framework developed by Braun and Clarke; a widely used approach for qualitative health research [36,37], drawing on principles of their thematic analysis and the Framework Method commonly used in multi-site health research.

## Applied thematic analysis

RFB engaged in repeated reading of the transcripts and field notes to become familiarized with the content, document initial impressions, analytic memos, and emergent ideas. This immersion provided the foundation for subsequent coding and theme development in relation to the study objectives. The transcriptions and notes were reviewed to identify and label key themes, ideas, and patterns that emerged from the discussions. Transcript data was de-identified by Adachi-Odai Solutions, removing names, personal identifiers and specific details (e.g., changing exact location of University of Alabama to a University in the US) and analyzed alongside moderator notes to support contextual interpretation of the data.

Coding was conducted inductively using a consensus-based approach, with analytic decisions discussed among the research team to support the consistent interpretation of codes and their application across transcripts, rather than relying on statistical intercoder reliability [37–39]. Codes were then categorized into broader thematic categories. This process involved grouping related codes together to identify overarching themes that represented patterned responses across participants and focus groups [36,39]. Themes were refined through iterative review, defined and named by RFB at this stage, and used as analytic constructs to guide the interpretation of the findings moving forward. This analytic approach reflects an applied thematic method commonly used in multi-site health research to support cross-group comparison and transparency in interpretation of the data rather than only formal measures of intercoder reliability.

## Cross-group synthesis

Data from all ten focus groups were compared and contrasted to identify commonalities and differences across the sites and participant groups, consistent with framework-based cross-case synthesis methods [40,41]. This cross-group synthesis helped in understanding the diversity of perspectives and experiences within the returning citizen population.

The notes taken during the focus groups were used to supplement and corroborate the findings from the transcriptions, providing a more holistic view of the data.

## Identification of key insights

The final step involved distilling the data into key insights and takeaways that answered the research objectives. This included identifying patterns, trends, and unique viewpoints that provided a deep understanding of the returning citizens' interactions with and attitudes towards research and science. Quotes from participants were adapted to maintain the grammatical integrity and remove identifiable linguistic features such as regional or accented speech. At the conclusion of the analysis, focus group recordings, transcripts and notes were destroyed to ensure protection of privacy for the participants.

## Ethics statement

This research was conducted in accordance with all relevant ethical guidelines and was deemed exempt by the All of Us Institutional Review Board (AoU IRB). The AoU IRB determined that it meets the criteria for exemption under 45 CFR 46.104(d)(2)(iii) which requires Limited IRB Review. The AoU IRB also determined that the project meets the requirements of 45 CFR 46.111(a)(7) as there are adequate provisions to protect the privacy of subjects and to maintain the confidentiality of data. The AoU IRB Protocol Number is 2023-01-CA-004. Verbal consent was obtained and no identifying information was collected.

## Results

### Participants

Ten focus groups were completed over five months, with a total of 67 participants across five states. The focus groups included a majority male participants at 72%. Nearly half (49%) of participants were between the ages of 26 and 40, then the next largest group (27%) were between 41 and 54 years old. Participants aged 18–25 and over 55 were nearly equally represented at 12% and 13% respectively. The largest racial group that participated identified as Black at 63%, following Hispanic and White individuals at 15% and 12%. From an economic perspective, 36% of participants had an annual household income between $10,000 and $24,999, 21% made over $50,000 with lower wage ranges representing 12–16% of participants (See Table 2). Below, participant quotes are attributed based on their partner, focus group, and participant within that focus group, using the format P#-FG#-P#.

### Focus group results

**Perspectives on research.** For our first objective to explore the perspectives of participants on research and science, there was alignment across participants that the goals should be to improve quality of life for themselves and society. When asked about what would contribute to this, participants shared that their physical and mental health were important to them, as well as understanding their ancestry and how it impacts their family. Older participants held more concerns around safety for research participation.

A large majority (70%) of the participants responded that research should lead to an improvement in quality of life. Many saw research as a means to develop treatments and improve healthcare outcomes, with an understanding that solutions are often not one-size-fits-all. For example, participant P1-FG1-P3 stated: "I think health research, number one, should achieve a kind of improvement in the previous rediscovery of health…and then we help eradicate a lot of health imbalances in society." Another participant P2-FG2-P1 spoke to the importance of diversity: "It's just important to make sure that you gather as much data from as many socioeconomic groups as possible and to do it over a long period of time."

A majority of participants, particularly aged 40 and older, shared a concern that research could treat them as guinea pigs and lab rats. Reflecting concerns about past transgressions, participant P4-FG1-P5 noted: "I think of lab rats, I think of guinea pigs. Anytime in my mind doing medical research, you're always tweaking that to make it have desired effects. But the first effects are going to be the most harmful." Another concern, expressed by participant P1-FG2-P3 was in reference to the Tuskegee Syphalis Study [42], "All the way back to the Tuskegee research, [research] always triggers. From lived experience, historical experience, it's been something that we took caution with. We're the last ones to want to try anything experimental or anything like that." Participants under 40 generally held more positive views on research including investigating and solving problems, and improvements for society such as participant P3-FG1-P2, "Trying to address some specific problems or challenges that could have practical solutions."

**The importance of health.** When asked, all participants stated that physical health is important. In addition, 90% of participants were interested in their health risk factors. Expressing an interest in learning more about their health, participant P4-FG1-P2 shared: "And I want to know more about it. …they…diagnose you with whatever they diagnose with. … They give you a handout…Well stop and explain that stuff to me. Let's talk about this for a second.." There was also a consensus that mental health was important; many participants faced mental health challenges while incarcerated. Discussing a lack of mental health care, participant P5-FG1-P4 stated: "You know, especially after being incarcerated, or even men of color. It's been taboo for a long time to deal with certain situations and issues, not understanding that it was mental health."With consideration for family health history and risk factors, 85% were interested in learning about their genetic ancestry. Participant P3-FG1-P2 stated: "I think that knowing your ancestry is very important, especially in the conversation that we've been having about knowing your health history. If you're connected with your ancestors, you can kind of beat some of the medical issues before they come up."

**Removing barriers to research participation: the role of community support.** For our second objective to identify facilitators and barriers to research participation, participants noted that appropriate compensation and support from community and/or family is crucial to being able to participate in activities beyond their top priorities. Sixty-six percent of participants felt supported by their community, and those who felt supported were more open to research participation. Speaking to their existing community support, participant P1-FG1-P2 shared: "I guess it made me more confident to participate because I would have good advisors around me. And I know that at the end of the day, if the people you're doing research with are good people, then yeah, it's going to help." Conversely, participant P3-FG2-P3 touched on the dangers of not having community support: "The child that's not embraced will burn the village down. … [If} You also have the resources; whether that's internet [access], a computer, to have the ability to participate in [research]."

Community engagement could be a significant factor in facilitating involvement in research. Participant P5-FG2-P5 discussed the level of comfort they experienced when research involves a known advocacy organization: "If the research is being conducted by an advocacy organization or an organization that is interested in implementing change, then, of course, I'm amenable to participating in such things.… We want to get data on this, on how further marginalizing and denigrating the reentry population not only harms that population but harms the fabric of our society as a whole, then I absolutely want to be involved with that".

**Making time and providing compensation.** When asked how participants spend their time. The most common answers were work (43%), health (15%), and family (13%); there were no significant differences when accounting for age, race or gender of the participant groups. These activities were prioritized over research participation and weighed against the potential time commitments to and compensation from research activities. Participant P4-FG1-P4 noted that flexibility in the time to complete research activities could improve uptake: "I've been approached to participate in research, it is so difficult for me because the [research appointments] are on days and times that I have to be at work." Most participants (59%) had previously participated in research, perhaps because they were connected with an advocacy organization. The most common reason for participating in research was a financial incentive, followed by personal benefits and curiosity. Participants cited a lack of opportunity as their primary reason for not participating in research. For example, participant P3-FG1-P3 noted: "I've spent all my adulthood incarcerated. So I haven't been available."

Taken together, participants relayed an interest in participating in research, often with the goal of helping others with experiences similar to theirs, and desire to overcome the logistical barriers that often prevent them from participating. Additional focus group analyses are available in S3 Text.

## Discussion

We conducted focus groups with the goal of learning from justice-impacted individuals about their interest in participating in research and any barriers to doing so. Our focus groups also revealed a desire to participate in research, in particular with the potential to positively impact them and the broader justice-impacted community. Similar motivators to contribute to society, gaining knowledge and healthcare, acquiring incentives, and obtaining social support, have been found in other studies as well as the potential for undue influence or coercion [43]. This work is in alignment with literature on the importance of including lived experience to achieve ethical incarceration-related research [44] and re-framing research from corrections and recidivism research to well being [45] and inclusion of the voices of formerly incarcerated individuals in research for human-centered outcomes. For example, a study of incarcerated women's experiences with participating in research on victimization found that the women were overwhelmingly positive about participating in the study and viewed it as an opportunity to share their story, reflect and help others [46].

To enable this group's participation, it is crucial for researchers to accommodate their prior commitments and offer adequate compensation. Notably, a recent review of all studies evaluating any incarcerated-based research found that 69% did not offer reimbursement to the incarcerated individuals [47]. To our knowledge, there is no existing literature focused on research participation compensation for formerly-incarcerated individuals.

Community-based participatory research has a long history of improving health research quality [48]. Collaboration with advocacy organizations that are supporting justice-impacted individuals can help studies ensure cultural competency and garner trust. This is consistent with the broader literature on utilizing community-based approaches to strengthen research [48,49]. Since participants had an existing relationship with an advocacy organization, it is possible that these findings may not scale to the general justice-impacted community, particularly if partnership with advocacy organizations is not included.

These findings can be applied to any research study that seeks to diversify perspectives and address significant health disparities. This study demonstrates the importance of including the participation of justice-impacted individuals in research studies. We recommend that future work explores feasibility and implementation of meaningful inclusion of justice-impacted individuals.

## Supporting information

**S1 Table. Recruitment Checklist.** This checklist was given to advocacy partners as an online survey to aid in their recruitment for eligibility and collect the demographic data of potential participants.
(DOCX)

**S1 Text. Recruitment Communications.** This section includes all recruitment communications to participants.
(DOCX)

**S2 Text. Focus Group Script.** This contains the IRB approved script used to guide focus group discussions.
(DOCX)

**S3 Text. More Detailed Focus Group Analysis.** This section provides more details and quotes from the focus group participants that aided the analysis of the results.
(DOCX)

## Acknowledgments

We are grateful to all research participants for sharing their experiences, and to all engagement partners for the work they do within and beyond this study.

## Author contributions

**Conceptualization:** Romina Foster-Bonds, Julia Moore Vogel.

**Data curation:** Romina Foster-Bonds.

**Formal analysis:** Romina Foster-Bonds.

**Funding acquisition:** Julia Moore Vogel.

**Methodology:** Romina Foster-Bonds.

**Project administration:** Romina Foster-Bonds.

**Supervision:** Julia Moore Vogel.

**Visualization:** Romina Foster-Bonds.

**Writing – original draft:** Romina Foster-Bonds.

**Writing – review & editing:** Romina Foster-Bonds, Julia Moore Vogel.

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
