## [Decision Letter · Decision Letter 0]

23 May 2025

PGPH-D-25-00414

Re-imagining health research to include the voices of justice-impacted individuals

Dear Dr. Vogel,

Thank you for submitting your manuscript to PLOS Global Public Health. After careful consideration, we feel that it has merit but does not fully meet PLOS Global Public Health’s publication criteria as it currently stands. Therefore, we invite you to submit a revised version of the manuscript that addresses the points raised during the review process.

EDITOR:

The manuscript has been evaluated by one reviewer, and their comments are available below.

The reviewer raised several concerns. The reviewer requests improvements to the reporting of methodological aspects of the study, for example, regarding more information on how the data collection for the focus group was completed. As a qualitative study, essential details from the COREQ checklist would address these concerns. Additionally, the reviewer would like a more robust, qualitative description of the results. The reviewer would like to see a more through connection between the results and the existing literature of the field in your discussion.

Could you please carefully revise the manuscript to address all comments raised?

Please note that we have only been able to secure a single reviewer to assess your manuscript. We are issuing a decision on your manuscript at this point to prevent further delays in the evaluation of your manuscript. Please be aware that the editor who handles your revised manuscript might find it necessary to invite additional reviewers to assess this work once the revised manuscript is submitted. However, we will aim to proceed on the basis of this single review if possible.

We look forward to receiving your revised manuscript.

Kind regards,

Katherine Demi Kokkinias, Ph.D.

Staff Editor

Journal Requirements:

Additional Editor Comments (if provided):

Reviewers' comments:

Reviewer's Responses to Questions

**Comments to the Author**

1. Does this manuscript meet PLOS Global Public Health’s publication criteria?

Reviewer #1: Partly

2. Has the statistical analysis been performed appropriately and rigorously?

Reviewer #1: I don't know

3. Have the authors made all data underlying the findings in their manuscript fully available (please refer to the Data Availability Statement at the start of the manuscript PDF file)?

Reviewer #1: Yes

4. Is the manuscript presented in an intelligible fashion and written in standard English?

Reviewer #1: Yes

Reviewer #1: Review Summary:

This article tackles a crucial and timely topic, emphasizing the importance of including justice-involved individuals in research and identifying barriers and facilitators to their participation. However, significant revisions are necessary before it is ready for publication. The methods, results, and discussion sections are currently underdeveloped. Given the importance of the topic, it deserves a thorough and thoughtful revision to reach its full potential.

Major Issues

1. Methods:

a. The methods section should include more detailed information. For example, a detailed description of the data collection process is missing. This should encompass details like who conducted the focus groups, the questions asked, and other relevant procedures.

b. For formerly incarcerated individuals, especially those who were incarcerated for long periods, using Zoom or smartphones can be challenging and may act as an additional barrier to participating in research. It should be clarified if staff assisted participants in using Zoom or smartphones/tablets, or if individuals with limited digital literacy were excluded from the study.

c. Since this is a qualitative study, the manuscript should adhere to the COREQ (Consolidated Criteria for Reporting Qualitative Research) checklist. This checklist provides guidance on the essential details to include in the methods section of the paper and the COREQ checklist should be included as a supplement.

2. Analysis:

a. The analysis section should be written in a narrative format rather than using bullet points to list the analytical steps taken.

b. The manuscript should also describe who participated in the analysis process.

3. Results:

a. Participant demographics should be detailed within the text of the results section.

b. Although the data analysis section describes a robust analysis, this is not adequately reflected in the results section. The results section lacks comprehensive details of the thematic analysis results and tends to offer a more quantitative than qualitative description.

4. Discussion:

a. The discussion should integrate how this study builds on and connects to existing research on this topic, including relevant discussions and citations of prior studies. This should include methods such as community-based participatory research that focus on the inclusion of justice-involved individuals in research. Furthermore, it should explore strategies to ensure greater participation from this population in all types of research.

Minor Issues

1. The introduction requires revisions to improve its flow and ensure it addresses the issues most relevant to the study. For instance, the second paragraph seems out of place and introduces the term "Returning Citizens," which is not used consistently throughout the manuscript (it reappears only on page 6). Multiple terms are used for the population of interest, which may cause confusion.

2. Paragraph 4 should be reorganized to maintain a consistent focus, avoiding abrupt shifts from one topic to another and then back again. Ensuring a logical and cohesive structure will enhance readability and comprehension.

**Do you want your identity to be public for this peer review?** For information about this choice, including consent withdrawal, please see our Privacy Policy

Reviewer #1: No

---

## [Decision Letter · Decision Letter 1]

3 Nov 2025

PGPH-D-25-00414R1

Re-imagining health research to include the voices of justice-impacted individuals

Dear Dr. Vogel,

Thank you for submitting your manuscript to PLOS Global Public Health. After careful consideration, we feel that it has merit but does not fully meet PLOS Global Public Health’s publication criteria as it currently stands. Therefore, we invite you to submit a revised version of the manuscript that addresses the points raised during the review process.

The manuscript has been evaluated by one reviewer, and their comments are available below.

The reviewer raised a number of concerns that need attention. They request additional information on methodological aspects of the study (such as the thematic analysis and cross-group synthesis), as well as some improvements in clarity.

Could you please revise the manuscript to carefully address the concerns raised?

We look forward to receiving your revised manuscript.

Kind regards,

Jen Edwards

Staff Editor

Journal Requirements:

Additional Editor Comments (if provided):

Reviewers' comments:

Reviewer's Responses to Questions

**Comments to the Author**

Reviewer #2: (No Response)

publication criteria?

Reviewer #2: Partly

3. Has the statistical analysis been performed appropriately and rigorously?

Reviewer #2: N/A

4. Have the authors made all data underlying the findings in their manuscript fully available (please refer to the Data Availability Statement at the start of the manuscript PDF file)?

Reviewer #2: Yes

5. Is the manuscript presented in an intelligible fashion and written in standard English?

Reviewer #2: Yes

Reviewer #2: This is an interesting project, on an important and interesting topic – capturing the voice of individuals who ordinarily lack a platform to be heard – this is commendable! However, I hope my comments below, help to strengthen this work further.

General style and points

The paper could benefit from improving conciseness in places – e.g., in the study design section ‘‘The information shared about the moderator was their name, that they were moderating for the respective Advocacy Organization and that the study was being conducted by Scripps Research’ – maybe reword to something like ‘Participants were shared the following information: moderators name, their role, and that Scripps Research was conducting the study’. It would be worth reading the paper with a similar approach to other sections too.

Pg5 there is a double ‘..’ after ‘(see Table 1 Advocacy Organization Partners)..’ – this should be singular

Ethics

The recruitment section suggests that participants were recruited by advocacy partners via phone calls – this seems a little targeted? It would be worth demonstrating how this was still on a voluntary, fully-informed, basis – this seems to stand out from the other methods e.g., text messages, social media posts – where people can choose / volunteer to get involved. Maybe explanation of this is required.

Data analysis

It appears that a form of Thematic Analysis was used – but it’s not clear which model or approach? The description of the process seems a little confused. You state that ‘transcripts were reviewed to identify and label key themes, ideas, and patterns’ but surely you start with codes – which you seem to say in the latter sentence. You also state ‘themes, ideas, and patterns that emerge’ which is not in keeping with up to date thinking about Thematic Analysis. This section needs some reworking, and I’d suggest using citations and references to demonstrate the underpinnings of the approach

Cross-group synthesis

Again it is unclear where this approach to analysis has come from, citations and references to this method of analysis would be useful. Whilst I appreciate this is a synthesis the compare and contrast across groups sits less comfortably with qualitative work and philosophy – providing citations and references would help create a more robust explanation.

**Do you want your identity to be public for this peer review?** For information about this choice, including consent withdrawal, please see our Privacy Policy

Reviewer #2: No

---

## [Decision Letter · Decision Letter 2]

5 Jan 2026

PGPH-D-25-00414R2

Re-imagining health research to include the voices of justice-impacted individuals

Dear Dr. Vogel,

Thank you for submitting your manuscript to PLOS Global Public Health. After careful consideration, we feel that it has merit but does not fully meet PLOS Global Public Health’s publication criteria as it currently stands. Therefore, we invite you to submit a revised version of the manuscript that addresses the points raised during the review process.

We look forward to receiving your revised manuscript.

Kind regards,

Baldeep Kaur Dhaliwal, PhD

Academic Editor

Journal Requirements:

**Additional Editor Comments (if provided):**

Please restructure your results. Your quotes should not be presented as bullets, and they should be better integrated into the content of your results. I also see that a previous reviewer made this comment, so please ensure that you take this step in your next revision - thanks!

**Reviewers' comments:**

Reviewer's Responses to Questions

**Comments to the Author**

Reviewer #3: All comments have been addressed

Reviewer #4: (No Response)

publication criteria?

Reviewer #3: Yes

Reviewer #4: (No Response)

3. Has the statistical analysis been performed appropriately and rigorously?

Reviewer #3: N/A

Reviewer #4: N/A

4. Have the authors made all data underlying the findings in their manuscript fully available (please refer to the Data Availability Statement at the start of the manuscript PDF file)?

Reviewer #3: No

Reviewer #4: (No Response)

5. Is the manuscript presented in an intelligible fashion and written in standard English?

Reviewer #3: Yes

Reviewer #4: Yes

**Reviewer #3:** I think that a plan on what the researchers are going to do after analyzing their findings in terms of policy changes, asking ethics bodies to make changes to make sure previously incarcerated people are not classified as vulnerable populations any more needs to be done for such a research to be of value to the community.

**Reviewer #4:** Thank you for this important piece of research! It is clearly an important topic, and I appreciate your work. Because this is another round of review and the work is important, I have been quite comprehensive in my recommendations. I hope that they will be useful in getting this piece successfully published.

General

For future articles, please provide line numbers to aid with review.

Your methods section is lengthy, and your TA approach is not coherent – it seems you actually used a codebook type approach (framework analysis or template analysis), but you have cited Braun and Clarke’s reflexive TA. Needs revision.

Your results sections is a little sparse and not always well-supported by quotes (for example, the last paragraph). This section should be prioritised for depth and word count – it is the purpose of this article!

Your discussion section needs to better link your own findings to the existing literature and research, showing what you add and its importance/relevance.

Abstract

Paragraph 2 – change ‘the nation’ to ‘the United States’.

Introduction

First paragraph and throughout – make sure ‘US’ is used after acronym introduced. See also final paragraph introduction.

Second paragraph

- “With growing incarceration, there are also significant disparities.” – this sentence is not clear/complete and could be improved. Consider: “With increasing incarceration, there are also growing inequities.” Or “While incarceration has increased throughout the US, these increases are not equitably distributed.”

- “Black men and women are five times more likely to be incarcerated than White men, and Hispanic men are 1.3 times more likely (8,9).” – not a complete sentence. Hispanic men are 1.3 times more likely than what/who? Could rephrase: “Black men and women are five times more likely, and Hispanic men are 1.3 times more likely, to be incarcerated than White men.

Para 3

- “There are severe, immediate and long standing racial and socioeconomic disparities with individuals who are incarcerated that negatively affect mental and physical health and significantly impair successful reintegration.” - not clear. It is not necessarily the disparity that affects physical/mental health, so much as being incarcerated which creates the disparity… Consider rephrasing: “There are severe, immediate and long-term racial and socioeconomic disparities for justice-impacted individuals compared to others, including negative effects on their physical, mental, and social health which often significantly impairs their successful societal reintegration.”

Para 4

- “The effects extend beyond the individual to family members” – improve for clarity. Consider: “The negative effects of being justice-impacted also extends beyond the individuals to their family members”.

- “45% of people in the US have an immediate family member who is incarcerated” – don’t start the sentence with a number; write out as “Forty-five percent”

Final para

- “Individuals with previous US justice system involvement are one of the largest and most underserved populations in the world.” – In the world? Or in the US? If the world, provide reference, as this seems unlikely.

Methods

Recruitment

– must state (if true) that participation was voluntary

- Include the number of recruited participants

Study design

- You’ve mentioned that there were 5 advocacy groups, but how many focus groups were there in total? Did each advocacy group have one focus group? How many people per focus group?

Risks and Mitigation

Para 2

- What is “PII”?

Data analysis

- The transcription section should appear under the data collection / study design heading.

- You need to say who did what part of the analysis, not just “the research team”. Use acronyms for your names. Who coded the transcripts? If more than 1 person, how did you code and compare with multiple coders? Did all authors sit together to categorize codes into broader themes?

- It does not seem accurate to say your coding was purely inductive, as it seems you came with some pre-determined objectives about facilitators and barriers to research participation?

- If you did use Braun and Clarke’s TA (called reflexive TA) then you need to acknowledge your positionalities and their effects on the results.

Cross-Group synthesis

- You have now cited framework analysis - a distinct method of thematic analysis from Braun and Clarke’s – that is a type of codebook analysis. I suggest engaging with Braun V, Clarke V. Toward good practice in thematic analysis: Avoiding common problems and be(com)ing a knowing researcher. International Journal of Transgender Health. 2023;24(1):1–6. DOI: 10.1080/26895269.2022.2129597 and Braun V, Clarke V. One size fits all? What counts as quality practice in (reflexive) thematic analysis? Qual Res Psychol. 2020;18(3):1–25. DOI: 10.1080/14780887.2020.1769238. It seems that your overall TA method was more aligned to codebook analysis?

Results

Participants

- Consistency in capitalising race terms

Focus group results

- “Each synopsis statement below is followed by quotes from the participants on that topic, adapted to maintain the grammatical integrity and remove identifiable linguistic features such as regional or accented speech:” – this has already been mentioned. Remove.

- Need theme headings, with results grouped accordingly.

- Need identifier/pseudonym/focus group number after each quote

- “85% were interested in learning about their ancestry.” – not so much ‘learning about their ancestry’ (which could mean a host of things including social and cultural), as learning about health risks inherited from their ancestors?

- The last paragraph is not supported with quotes.

- In general, more needs to be made of this section.

Discussion

- I recommend starting your discussion with a very brief reminder of your research aim/s, and what you found.

- The first paragraph is largely a repetition of what is said in the introduction and is not directly relevant to your findings.

- Second paragraph is good as it compares your findings to other similar research. Discussion needs to be more in this vein, showing where your findings are novel or similar.

- As above, you need to more specifically discuss your own findings in relation to the literature. For example, where you say: “Collaboration with advocacy organizations that are supporting justice-impacted individuals can help studies ensure cultural competency and garner trust.” – while this may be true, you have not mentioned cultural competency or trust of advocacy groups in your results, and appears to come from nowhere.

- Final sentence: “We hope this shows it is feasible to include the participation of justice-impacted individuals in research studies.” – your study is not about feasibility and does not prove it is feasible to include these individuals. It does show that it is important, however, and you offer recommendations for improving participation. Revise final sentence to better reflect your study.

Thank you.

**Do you want your identity to be public for this peer review?** For information about this choice, including consent withdrawal, please see our Privacy Policy

Reviewer #3: **Yes:** Anjum John

Reviewer #4: No

---

## [Decision Letter · Decision Letter 3]

8 Feb 2026

PGPH-D-25-00414R3

Re-imagining health research to include the voices of justice-impacted individuals

Dear Dr. Vogel,

Thank you for submitting your manuscript to PLOS Global Public Health. After careful consideration, we feel that it has merit but does not fully meet PLOS Global Public Health’s publication criteria as it currently stands. Therefore, we invite you to submit a revised version of the manuscript that addresses the points raised during the review process.

We look forward to receiving your revised manuscript.

Kind regards,

Baldeep Kaur Dhaliwal, PhD

Academic Editor

Journal Requirements:

Reviewers' comments:

Reviewer's Responses to Questions

**Comments to the Author**

Reviewer #4: All comments have been addressed

publication criteria?

Reviewer #4: Yes

3. Has the statistical analysis been performed appropriately and rigorously?

Reviewer #4: N/A

4. Have the authors made all data underlying the findings in their manuscript fully available (please refer to the Data Availability Statement at the start of the manuscript PDF file)?

Reviewer #4: (No Response)

5. Is the manuscript presented in an intelligible fashion and written in standard English?

Reviewer #4: Yes

**Reviewer #4: Well done on producing this manuscript. Minor recommendations: Move our inclusion criteria from study design to recruitment, and consider a brief summarising sentence of your results before your discussion section.**

**Do you want your identity to be public for this peer review?** For information about this choice, including consent withdrawal, please see our Privacy Policy

Reviewer #4: No

---

## [Editor Report · Decision Letter 4]

12 Feb 2026

Re-imagining health research to include the voices of justice-impacted individuals

PGPH-D-25-00414R4

Dear Dr. Vogel,

We are pleased to inform you that your manuscript 'Re-imagining health research to include the voices of justice-impacted individuals' has been provisionally accepted for publication in PLOS Global Public Health.

Best regards,

Baldeep Kaur Dhaliwal, PhD

Academic Editor
